

# Identification of genes and gene expression associated with dispersal capacity in the mountain pine beetle, *Dendroctonus ponderosae* Hopkins (Coleoptera: Curculionidae)

Victor A. Shegelski[1], Maya L. Evenden[1], Dezene P.W. Huber[2] and Felix A.H. Sperling[1]

[1] Department of Biological Sciences, University of Alberta, Edmonton, Alberta, Canada
[2] Faculty of Environment, University of Northern British Columbia, Prince George, British Columbia, Canada

## ABSTRACT

Dispersal flights by the mountain pine beetle have allowed range expansion and major damage to pine stands in western Canada. We asked what the genetic and transcriptional basis of mountain pine beetle dispersal capacity is. Using flight mills, RNA-seq and a targeted association study, we compared strong-flying, weak-flying, and non-flying female beetles from the recently colonized northern end of their range. Nearly 3,000 genes were differentially expressed between strong and weak flying beetles, while weak fliers and nonfliers did not significantly differ. The differentially expressed genes were mainly associated with lipid metabolism, muscle maintenance, oxidative stress response, detoxification, endocrine function, and flight behavior. Three variant loci, two in the coding region of genes, were significantly associated with flight capacity but these genes had no known functional link to flight. Several differentially expressed gene systems may be important for sustained flight, while other systems are downregulated during dispersal and likely to conserve energy before host colonization. The candidate genes and SNPs identified here will inform further studies and management of mountain pine beetle, as well as contribute to understanding the mechanisms of insect dispersal flights.

## INTRODUCTION

Flight-based dispersal in insects is influenced by many biotic, abiotic, temporal and spatial factors (*Stinner et al., 1983*; *Bowler & Benton, 2005*; *Jones et al., 2019*). Individual insects respond to these factors with physiological, morphological, and behavioral adaptations that determine their flight capabilities (*Roff & Fairbairn, 2007*; *Jones et al., 2019*). In many insects, flight and dispersal capacity may manifest as discrete differences between dispersal and non-dispersal types (*Zhang et al., 2019*), but, in others, such as mountain pine beetle, dispersal distance is a continuous trait (*Shegelski, Evenden & Sperling, 2019*).

Corresponding author
Victor A. Shegelski,
shegelsk@ualberta.ca

In another dispersing pest species, the cotton bollworm, *Helicoverpa armigera*, such continuous traits associate with large suites of genes (*Jones et al., 2015*). Disentangling the mechanisms that underlie flight capacity can provide valuable insights into the dispersal and spread of economically important insect species (*Kristensen, De Barro & Schellhorn, 2013*; *Portman et al., 2020*).

Mountain pine beetles, *Dendroctonus ponderosae* Hopkins (Coleoptera: Curculionidae: Scolytinae), contribute to the maintenance of healthy pine forests (*Safranyik & Carroll, 2006*), but can cause significant damage during major outbreaks (*Hart et al., 2015*). The most recent outbreak in western Canada breached the Northern Rocky Mountains (*de la Giroday, Carroll & Aukema, 2012*) and expanded the range of the mountain pine beetle to include a novel host tree, jack pine, *Pinus banksiana* Lamb. (*Cullingham et al., 2011*), which has a distribution that extends through the boreal forest toward further pine hosts in eastern Canada and USA.

Morphology is a significant indicator of flight capacity by mountain pine beetles (*Evenden, Whitehouse & Sykes, 2014*; *Shegelski, Evenden & Sperling, 2019*); however, many beetles with strong flight-related morphology actually fly very little, displaying a large amount of unexplained variation in flight performance (*Shegelski, Evenden & Sperling, 2019*). Genetic factors constitute a plausible component of this variation. Numerous genes that relate to other life phases and processes of mountain pine beetles have been identified, including pheromone biosynthesis (*Keeling et al., 2013a*; *Nadeau et al., 2017*), overwintering cold tolerance (*Bonnett et al., 2012*; *Robert et al., 2016*; *Fraser et al., 2017*), detoxification of host defenses and reproduction (*Robert et al., 2013*; *Pitt et al., 2014*; *Huber & Robert, 2016*). However, the dispersal phase of these beetles remains essentially unexplored from a genomic standpoint.

Our study aimed to identify genetic correlates of dispersal capacity in mountain pine beetles. Dispersal-related genes in other insects have been linked to regulation of metabolic rate (*Wheat et al., 2010*; *Jones et al., 2015*; *Zhou et al., 2020*), muscle tracheation (*Marden et al., 2013*), generally improved flight capability (*Niitepõald et al., 2009*; *Wheat et al., 2010*), and increased dispersal behavior (*Zhou et al., 2020*). We sought to determine what suites of genes are most associated with flight and dispersal-related functions in mountain pine beetles. To achieve this, we combined a computer-linked flight mill bioassay with differential gene expression analysis using RNA-Seq and a subsequent targeted association study. The identification of flight-related genes will allow better understanding of the mechanisms associated with insect flight and dispersal and potentially lead to more effective management of the spread of mountain pine beetle, in both its established and newly expanded ranges.

# METHODS

## Sample collection and preparation

Four lodgepole pine trees infested with mountain pine beetles were selected from sites near Grande Prairie, Alberta, Canada (three trees from Site 1 at 54.57 N, 119.42 W; and one tree from Site 2 at 54.19 N, 118.68 W) in October 2015. Two 50-cm bolts were cut from

each tree at about one m above ground level. The cut ends of the bolts were sealed with paraffin wax upon felling, and bolts were transported to a laboratory and stored at 4 °C for 6 months, to emulate winter conditions.

For beetle emergence, bolts were placed in separate 136 L opaque plastic emergence chambers at 24 ± 1 °C in April 2016, and emerged beetles were collected daily. Individual beetles were separated by sex based on the presence or absence of beetle stridulation at 24 ± 1 °C, using the methods of *Rosenberger, Venette & Aukema (2016)*. In order to reduce metabolic expenditures of beetles before flight, individual beetles were stored at 4 °C in two mL centrifuge tubes with a small piece of paper until the flight bioassay (*Evenden, Whitehouse & Sykes, 2014*), which was performed three to five days post-emergence.

## Flight mill bioassay and sample selection

Only female mountain pine beetles were used in this study, as females initiate host location in nature (*Blomquist et al., 2010*), and we wanted to control for potential effects of sex both in the bioassays and in subsequent analyses. Prior to flight bioassays, female beetles were weighed and measured at 24 ± 1 °C for a separate morphological analysis (*Shegelski, Evenden & Sperling, 2019*). Beetles were then attached to individual two cm long aluminum wire tethers with a diameter of 0.32 mm. LePage® Heavy Duty Contact Cement was used to adhere the tether to the pronotum of each individual, taking care to avoid interference with wing or elytral movement.

A total of 124 female beetles were flown on flight mills. Data was recorded on flight distance, duration (time spent in flight) and propensity (the number of times flight was initiated after a minimum 5 second period of no flight), using methods for computer-linked flight mill bioassays described by *Evenden, Whitehouse & Sykes (2014)*. The flight bioassays were 22 h long, which allowed a 2-h period for specimen processing. Conditions in the flight chamber consisted of a 16:8 light:dark photoperiod and temperature held at 22.5 °C. Flights began 2 h after the initiation of the photoperiod light phase in the chamber, giving specimens a total of 14 h of potential flight time in light, and 8 h in dark.

The seven beetles with the highest and seven with the lowest total flight distances were used to represent the strongest and the weakest fliers, respectively. In addition, four beetles that did not fly, but demonstrated vigor and a full range of motion in the elytra and wings, were selected to represent non-fliers. Selected beetles were flash frozen in liquid nitrogen after the flight bioassay and stored at −80 °C until RNA extraction. All other assayed beetles were stored at −20 °C in 85% EtOH until use. Three voucher specimens from the same location have been submitted to the E.H. Strickland Entomological Museum at the University of Alberta, Canada (Accession numbers UASM391992, UASM391993 and UASM391994).

Samples for the association study included 59 female beetles taken from the remaining specimens that were not used for other analyses. This included a total of 31 strong and 28 weak fliers that were randomly selected from the upper and lower quartiles of total flight distance.

## RNA extractions, RNA-Seq library preparation & sequencing

We used Zymo Research direct-zol RNA MiniPrep kits for total RNA extractions, with a DNase I treatment and Invitrogen TRIzol as a medium for specimen homogenization. RNA quality was checked using an Agilent 2100 Bioanalyzer system and quantified using Invitrogen Qubit RNA fluorometric quantification. cDNA synthesis and RNA-seq library preparation used an IlluminaTruSeq Stranded RNA LT Kit and followed the recommended protocol, which included poly-A selection for mRNA purification. Prior to sequencing, quality and quantity of cDNA was checked again using an Agilent 2100 Bioanalyzer. Library sequencing was performed on an Illumina NextSeq 500 platform at the Molecular Biology Service Unit (MBSU) at the University of Alberta. Sample groups were evenly split between two NextSeq 500 runs to provide an optimal number of reads.

## Sequence data mapping & differential expression analysis

Our experiment included 18 female beetles belonging to three flight phenotypes (seven strong fliers, seven weak fliers, and four non-fliers). The male mountain pine beetle genome (*Keeling et al., 2013b*) was used for alignment of sequence data as it contains 1,504 more genes than the female genome. This difference is largely due to the presence of Y chromosome fragments but may also include some autosomal genes of importance. Sequence data was aligned using Bowtie2 version 2.1.1.3 (*Langmead & Salzberg, 2012*) and Tophat2 version 2.1.1 (*Kim et al., 2013*), with a maximum of five alignment sites, and otherwise default settings. These data were then sorted and indexed using SAMtools version 1.5 (*Li et al., 2009*).

Differential expression analysis was performed using the R package DESeq2 (*Love, Huber & Anders, 2014*), in R version 3.3.3 (*R Core Team, 2017*). We chose to use DESeq2 for its stringency (*Rajkumar et al., 2015*), potentially causing more false negatives (Type II error) than false positives (Type I error). BAM files were read using RSamtools (*Morgan et al., 2017*) and annotated read count tables were produced using the packages GenomicFeatures and GenomicAlignments (*Lawrence et al., 2013*). Differentially expressed genes with a false discovery rate (FDR) < 0.01 were considered candidate genes related to flight. A principal component analysis (PCA) was performed on regularized logarithm-transformed gene expression data to visualize relationships between the specimens based on gene expression without bias towards highly expressed genes (*Love, Huber & Anders, 2014*).

## Enrichment analysis & KEGG pathway analysis

We used Blast2GO version 5.2.5 (*Götz et al., 2008*) to identify trends in gene ontology (GO) annotations and Kyoto Encyclopedia of Genes and Genomes (KEGG; *Kanehisa & Goto, 2000*) pathways for the differentially expressed gene transcripts. Nucleotide sequences of these transcripts were blasted to the NCBI nr database, retaining the three best hits and otherwise using default search parameters. Transcript sequence data was mapped and annotated with GO terms, then also annotated based on the Interpro database to validate GO annotations. Enrichment analysis using Fisher's exact test (FDR < 0.05) tested upregulated and downregulated genes separately for significant overrepresentation
of GO terms between the strong and weak flight phenotypes. KEGG pathway analysis was also performed on all differentially expressed transcripts to identify flight-related pathways.

### Association study: DNA extractions, sequencing & mapping

DNA extractions for 59 female beetles used a Qiagen DNEasy Blood & Tissue kit, following standard protocol with an optional RNAse A treatment and library preparation using the ddRAD protocol of *Peterson et al. (2012)*. Single-end sequencing was performed at the University of Alberta MBSU using the Illumina NextSeq500 platform. Initial data processing and quality checking of the raw sequence data followed protocols by *Campbell et al. (2017)*.

A common weakness of association studies is the large number of comparisons may diminish power (*Visscher et al., 2017*). To increase statistical power, we reduced the number of tested loci by focusing on candidate regions identified through our RNA-seq experiment. Alignment for the association study used the subset of scaffolds from the male mountain pine beetle draft genome (*Keeling et al., 2013b*) that contained one or more flight-related genes based on our differential gene expression analysis. This alignment used the BWT-SW algorithm in BWA version 0.7.17 (*Li & Durbin, 2009*), and alignment quality was checked using SAMtools version 1.9 (*Li et al., 2009*).

Stacks 2.0 ref_map.pl (*Rochette, Rivera-Colón & Catchen, 2019*) was used to identify single nucleotide polymorphism (SNP) sites. We allowed a minor allele frequency of 1%, and loci were initially retained if they were present in at least 80% of the 59 specimens (*Paris, Stevens & Catchen, 2017*). Further filtering used vcftools version 0.1.14 (*Danecek et al., 2011*), with reads with a genotype quality score below 30 and SNP sites with more than 2% missing data among individuals excluded from the final data set.

### Targeted association study

We used TASSEL version 5.2.54 (*Bradbury et al., 2007*) to perform an association study in which we tested SNP sites for associations with total flight distance as well as flight propensity, which is likely an important predictor of realized dispersal in nature (*Steyn, Mitchell & Terblanche, 2016*). For this study we used 31 strong and 28 weak fliers. SNP sites with heterozygosity in less than 5% or greater than 95% of the samples were filtered, as these sites may represent genotyping errors (*Leal, 2005*). We used identity by state to account for relatedness between individuals, a principal component analysis (PCA) to account for population structure and stratification, and a generalized linear model (GLM) with permutation testing (1,000 permutations) to account for non-normal distributions in the phenotypic data and FDR (*Che et al., 2014*). The permutated $p$-value (perm $p$) was used to determine significance at $\alpha = 0.05$.

## RESULTS

### Flight mill bioassay

A summary of the bioassay results can be seen in Table 1 and Fig. 1. On average, female beetles from all flight bioassays ($n = 124$) flew 7.4 km and initiated flight 176 times. Beetles

**Table 1 Summaries of flight and sequence data for each sample group.**

| Beetle Group | n | Average flight distance (km) | Minimum flight distance (km) | Maximum flight distance (km) | Average number of flights | Minimum number of flights | Maximum number of flights | Average number of reads (millions) (>Q30) | Minimum number of reads (millions) (>Q30) | Maximum number of reads (millions) (>Q30) |
|---|---|---|---|---|---|---|---|---|---|---|
| All flown beetles | 124 | 7.40 | 0.01 | 28.81 | 176.00 | 2 | 2,806 | – | – | – |
| Beetles used for RNA-Seq | | | | | | | | | | |
| All beetles | 14 | 10.08 | 0.04 | 28.81 | 72.80 | 2 | 327 | 65.57 | 25.17 | 108.83 |
| Weak fliers | 7 | 0.23 | 0.04 | 0.69 | 61.70 | 2 | 210 | 67.14 | 44.32 | 91.50 |
| Strong fliers | 7 | 19.90 | 9.90 | 28.81 | 83.90 | 5 | 327 | 61.48 | 25.17 | 77.15 |
| Non-fliers | 4 | 0.00 | 0.00 | 0.00 | 0.00 | 0 | 0 | 69.99 | 25.60 | 108.83 |
| Beetles used for flight association study | | | | | | | | | | |
| All beetles | 59 | 6.38 | 0.01 | 24.57 | 229.83 | 4 | 2,806 | 1.32 | 1.32 | 1.32 |
| Weak fliers | 28 | 838.31 | 10.36 | 3,666.26 | 164.36 | 4 | 1,968 | 1.30 | 0.68 | 2.17 |
| Strong fliers | 31 | 11,392.52 | 5,229.04 | 24,568.30 | 288.97 | 23 | 2,806 | 1.35 | 0.32 | 1.73 |

specifically used for the differential expression analysis ($n = 14$, excluding non-fliers) flew an average of 10.08 km and initiated flight an average of 72.8 times. Beetles selected to represent the strong flight phenotype had an average flight distance of 19.9 km and initiated flight an average of 83.9 times, while the group representing the weak flight phenotype flew 0.231 km on average and initiated flight an average of 61.7 times. There was a significant difference in total flight distance between strong ($n = 7$) and weak ($n = 7$) flight phenotypes (t stat = 8.2, df = 6, $P < 0.0001$), but there was no significant difference in flight propensity (the number of times flight was initiated) (t stat = −0.45, df = 10, $p = 0.333$). Although the majority (88.5%) of flight by these beetles took place during the light period, beetles with a strong flight capacity flew significantly more in the dark than did weak fliers (t stat = 2.253, df = 12, $p = 0.0218$; Fig. 2). On average, strong fliers flew 84.4% of their total flight time in the light, and 15.6% in the dark. This compared to weak fliers that on average flew 95.6% of their total flight time in the light, and 4.4% in the dark.

The 59 female beetles used for the association study flew an average of 6.4 km and initiated flight an average of 229.8 times. The strong fliers used for the association study flew significantly farther than the weak fliers (t stat = 12.3, df = 57, $p < 0.001$). The strong flight beetles in this data set ($n = 31$) flew an average of 11.4 km, and the weak fliers ($n = 28$) flew an average of 0.8 km. There was no significant difference in flight propensity, however, between the strong and weak fliers used for the association study (t stat = 1.1, df = 57, $p = 0.297$) (Table 1).

### Differential expression analysis

A total of 1.735 billion reads were sequenced *via* RNA-Seq, and 1.283 billion reads were > Q30. On average, there were 65.5 million reads per sample, and this ranged from

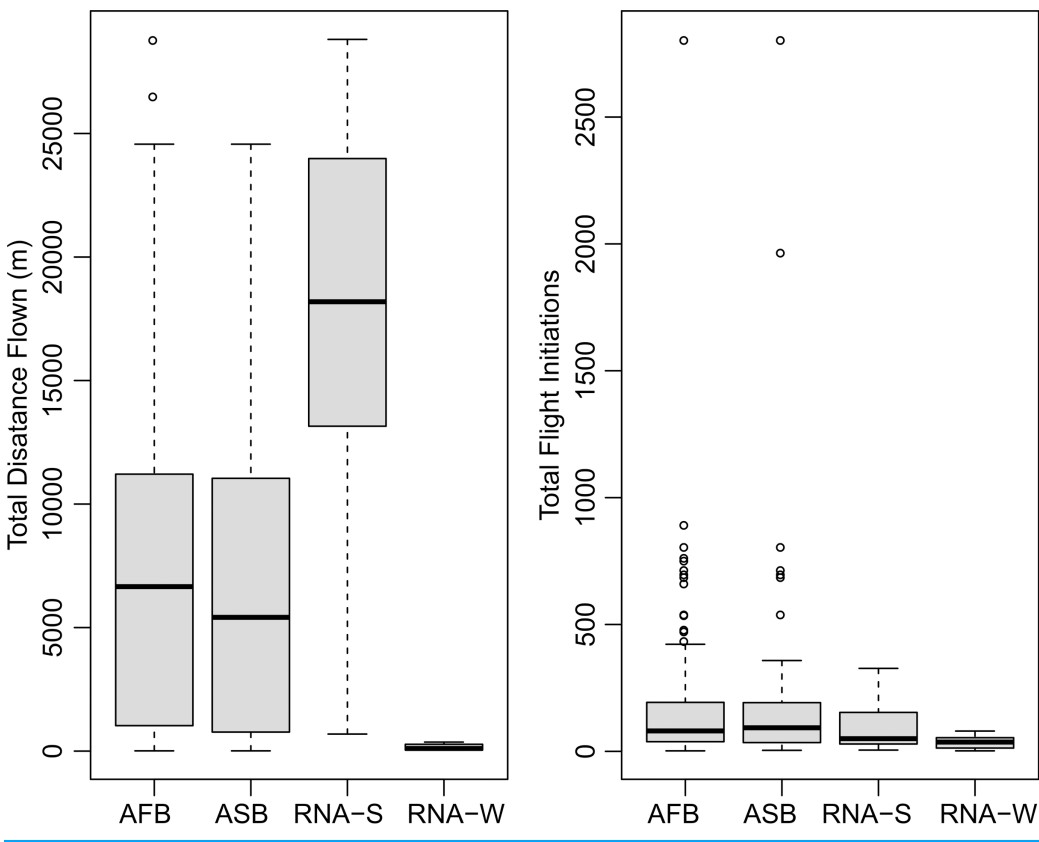

**Figure 1 Summary of flight statistics from flight mill bioassays.** Groups represented include all flown beetles (AFB), association study beetles (ASB), RNA-Seq experiment beetles that demonstrated strong flight (RNA-S), and RNA-Seq beetles that demonstrated weak flight (RNA-W).

25.2 million reads to 108.8 million reads (Table 1). Principal component analysis of variation in gene expression by the three phenotypes (strong, weak & non-fliers) showed a clear separation of strong fliers from weak and non-fliers. It also showed substantial overlap between non-fliers and weak fliers, with weak fliers having the highest variation in gene expression (Fig. 3A). No genes were significantly differentially expressed between weak fliers and non-fliers (Fig. 3B), and further analyses focused on the differences between strong and weak fliers. The list of differentially expressed genes between strong and non-fliers is in Table S1.

In comparisons of strong to weak fliers, differential expression analysis using DESeq2 revealed 2,741 differentially expressed genes (Fig. 3C; Table S1). Of these, 1,486 (54.2%) of the genes were upregulated and 1,255 (45.8%) were downregulated. Of the differentially expressed genes, 387 were uncharacterized. In this study, we focus on genes that could be broadly categorized as relating to metabolism (resource consumption and energy production), muscle form and function (physical structure and components of flight muscle), oxidative stress and detoxification (ability to remove metabolic wastes during flight), the endocrine system (regulation of flight-related physiological and behavioral

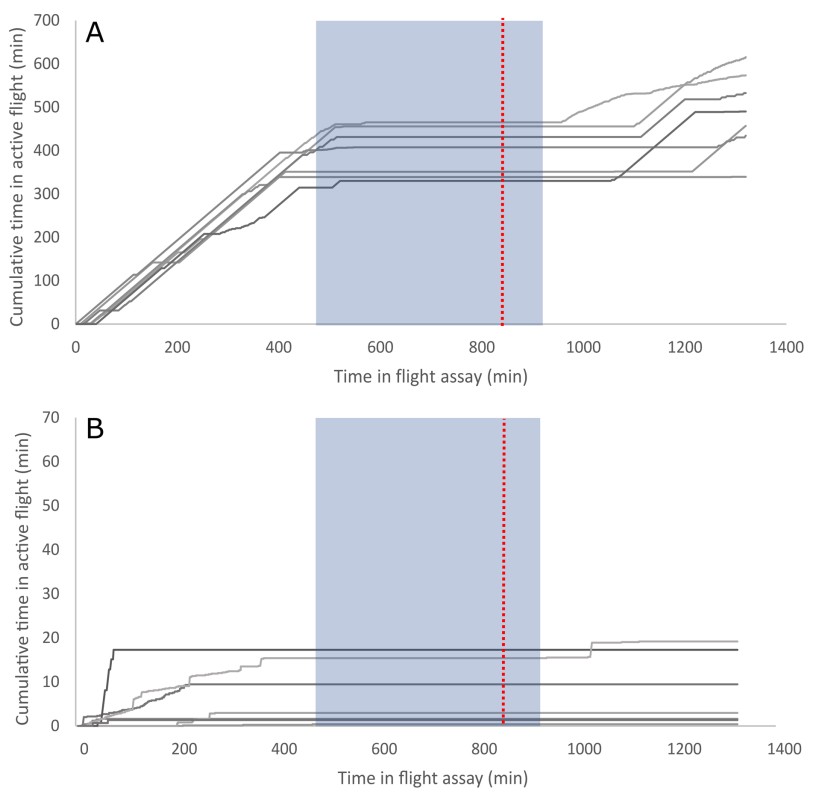

**Figure 2 Individual flight patterns of strong fliers (A) and weak fliers (B).** The dashed red line indicates when lights were switched off in the flight room to simulate night, and the shaded box indicates actual night based on June 20th (dusk at 22:00, dawn at 05:00). Note that the scale of the Y-axis is 10× less in (B), the weak flier graph, to allow for observable flight patterns.

mechanisms), and flight behavior (those affecting innate response and inclination relating to flight) (summarized in Table S2).

We identified 20 genes related to various stages of lipid metabolism, and all but three were upregulated with flight (Table S2). The majority were lipases, although there were also reductases and dehydrogenases, among others.

We found 15 differentially expressed genes relating to muscle form and function (Table S2), including four collagen alpha chains (all upregulated with flight), and five related to myosin (two upregulated with flight and three downregulated), including paramyosin, which is structurally integral to indirect flight muscle (*Liu et al., 2003*). We also found six differentially expressed myotubularin transcripts (three upregulated with flight and three downregulated with flight), which relate to muscle maintenance (*Laporte et al., 2001*).

We identified 10 differentially expressed genes involved in oxidative stress management (Table S2), including five glutathione S-transferases (GSTs; one theta, three sigma, one ambiguous Delta/Epsilon), one microsomal GST, and four antioxidants (two phospholipid hydroperoxide glutathione peroxidases, and two peroxiredoxins). We also found 23 differentially expressed cytochrome P450 genes, including seven CYP4s, nine CYP6s, three
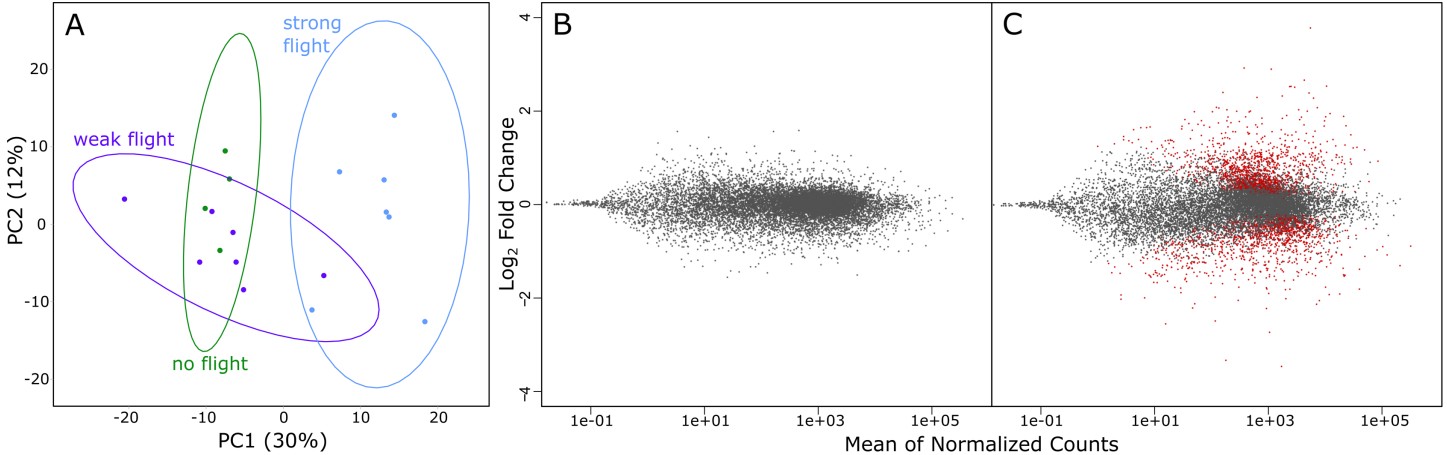

**Figure 3** (A) rlog principal component analysis of variation in gene expression in the three flight groups. Ellipses show 95% confidence intervals. (B) The log ratio vs mean average plot (MA plot) shows differential expression analysis between weak and non-fliers. There were no genes significantly differentially expressed with flight at α = 0.01. (C) MA plot shows differential expression analysis between strong and weak fliers. Significant differentially expressed genes with flight (α = 0.01) are shown in red.

CYP9s, two CYP307s, one CYP 302, and one CYP 28. Also related to stress response, we identified seven heat shock protein transcripts that were differentially expressed.

There were 10 transcripts related to endocrine function, including four related to juvenile hormone (JH) and ecdysone systems, and six related to insulin and insulin-like growth factor (IGF; Table S2). Both JH-related genes were upregulated; these both coded for juvenile hormone epoxide hydrolase (JHEH), which deactivates JH. There was upregulation of ecdysone 20-monooxygenase, which catalyzes production of the hormone 20-hydroxyecdysone (20E), and the ecdysone-inducible protein E75, an orphan hormone receptor (*Segraves & Hogness, 1990*), was downregulated. Insulin is involved in metabolism, growth, and development (*Nässel, Liu & Luo, 2015*). Of the six insulin family-related genes, there were four upregulated receptors (two insulin receptors and two insulin-like growth factor receptors), one downregulated insulin-degrading enzyme, and one downregulated insulin-like growth factor-binding protein.

Some potential behavioral genes that were differentially expressed included two isoforms of nocturnin, which relates to light-mediated behavioral response, a circadian clock-controlled protein, and protein alan shepard, which is related to gravitaxis (*Armstrong et al., 2006*; Table S2).

Nine differentially expressed transcripts were related to olfaction (Table S2). Of these, transcripts for two chemosensory proteins, three odorant receptors, and two odorant-binding proteins were downregulated with flight, and two odorant binding-protein transcripts were upregulated with flight.

## GO enrichment analysis

Of the 2,741 differentially expressed gene transcripts, 2,556 (93.3%) of the transcripts had InterPro hits, and 1,358 (49.5%) had one or more GO annotations. Enrichment analysis of upregulated GO terms was performed in two categories–biological process and molecular function. The 100 significantly overrepresented GO terms (89 biological

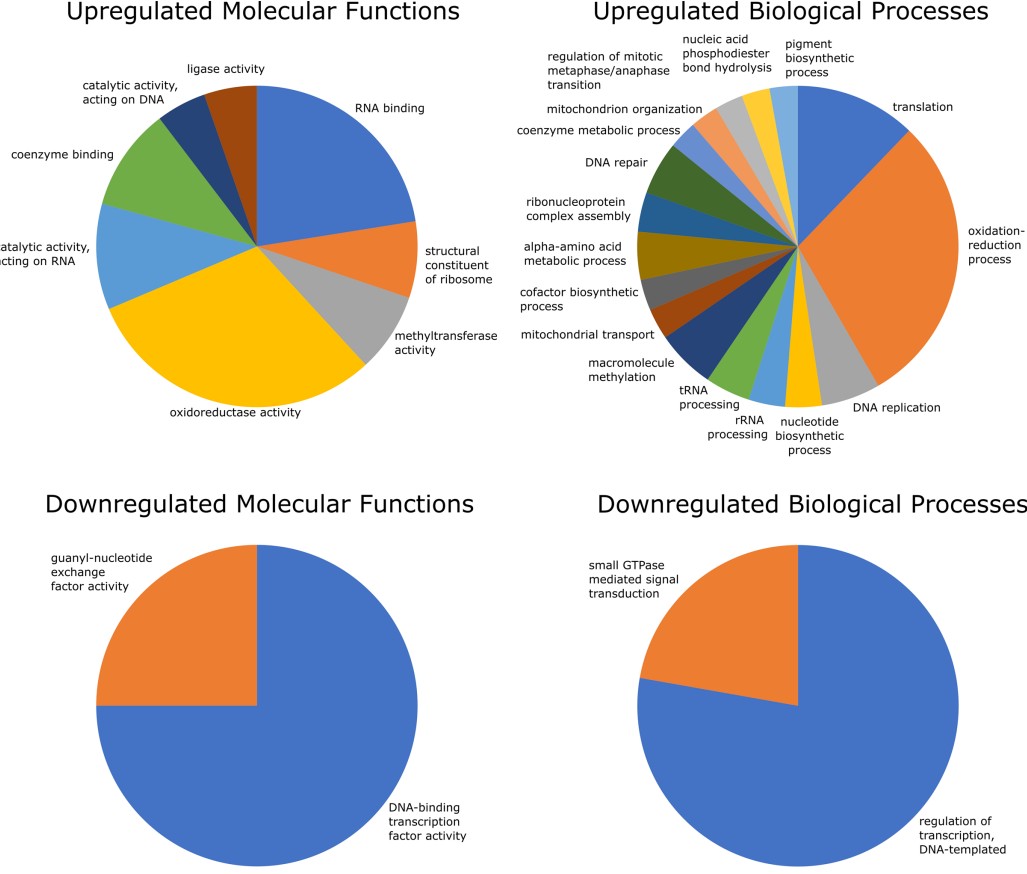

**Figure 4 GO (Gene Ontology) enrichment analysis reduced to most specific terms where possible.** All GO annotations shown were significantly different between strong and weak fliers (FDR = 0.05). Pie charts show proportions represented in the total number of sequences relating to differentially expressed GO terms.

processes and 11 molecular functions; FDR = 0.05), were condensed to 25 related GO terms with more specific functions. These included eight molecular functions and 17 biological processes relating to oxidoreductase activity, coenzyme binding, oxidation-reduction processes, mitochondrial transport and organization, and coenzyme metabolic processes (Fig. 4).

Enrichment analysis of the downregulated GO terms revealed 28 overrepresented terms (24 biological processes and four molecular functions; FDR = 0.05), which were reduced to four more specific GO terms, including two molecular functions and two biological processes related to DNA transcription and signal transduction (Fig. 4).

## KEGG pathways

KEGG pathway analysis of all differentially expressed genes revealed 52 pathways represented by at least two enzymes, 42 of which were represented by at least three enzymes (Fig. 5). Several metabolic pathways included: the metabolism of glycerolipids, purine, pyruvate, and sphingolipids, pantothenate and coenzyme A biosynthesis, glycolysis, and the citric acid cycle. Detoxification and stress response were also represented in the KEGG
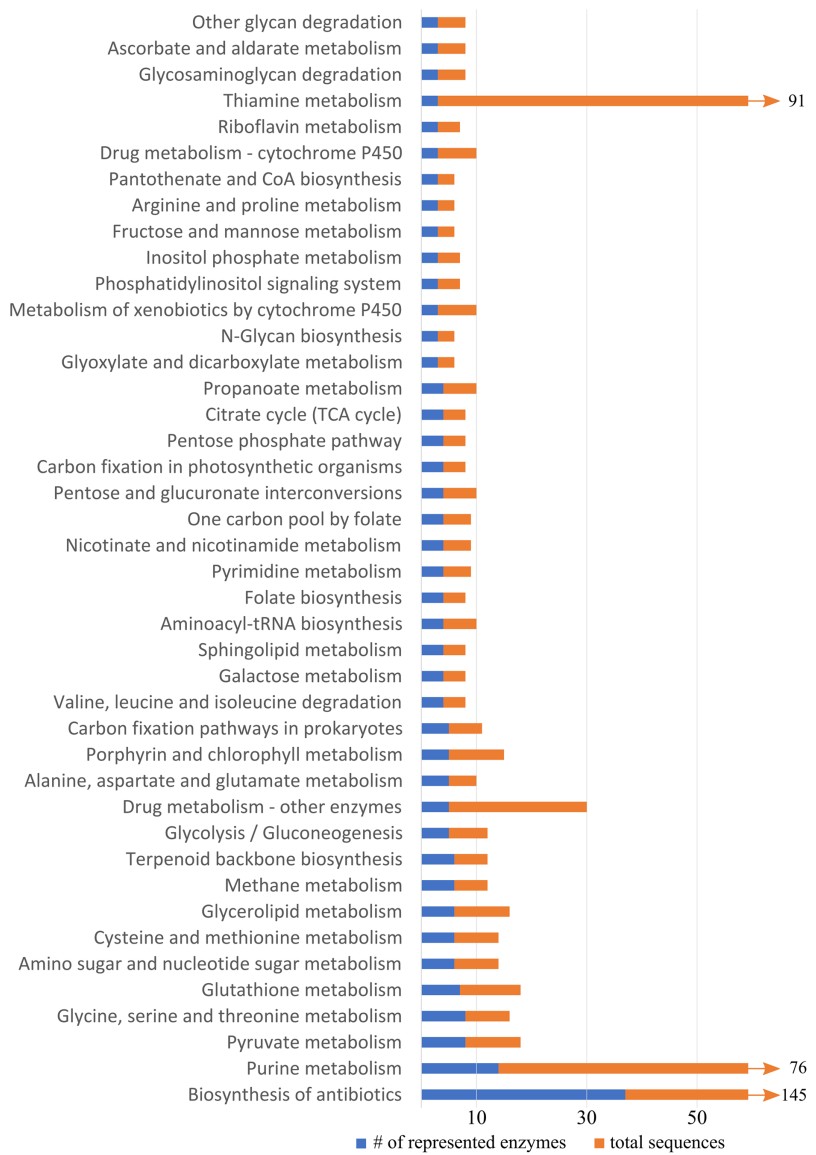

**Figure 5** Representation of KEGG (Kyoto Encyclopedia of Genes and Genomes) pathways with more than three differentially expressed genes. Blue shows number of enzymes of a pathway represented in our data, and orange shows the total number of sequences related to enzymes in the pathway. Pathways with >60 sequences have been truncated.

pathways, including detoxification, biosynthesis of antibiotics, drug metabolism, and metabolism of xenobiotics by cytochromes P450. We also found pathways associated with glutathione metabolism, which may relate to oxidative stress. The steroid hormone biosynthesis pathway, which may relate to hormone-mediated flight behaviors, was also represented, but only by three sequences that all coded for the same enzyme.

## Targeted association study

We identified 714 scaffolds containing differentially expressed flight genes. Of these, 347 scaffolds contained no SNPs, and the remaining 367 scaffolds contained a total of

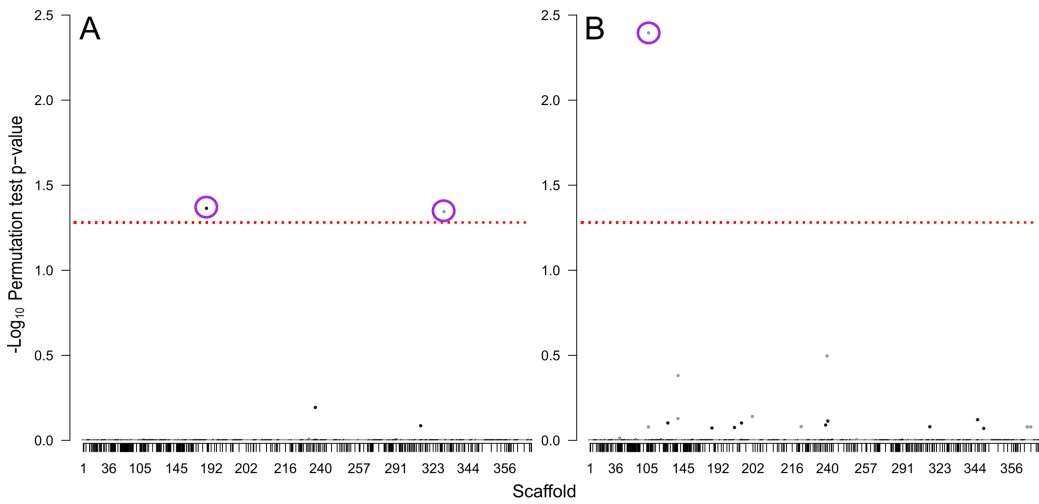

**Figure 6 Manhattan plots showing −log$_{10}$ permutation test $p$-values for each SNP based on testing for association with (A) total distance flown and (B) the total number of times flight was initiated.** SNPs are shaded by scaffold, alternating between black and white, the dashed red line indicates the threshold of significance based on a permutation test $p$-value of 0.05, and purple circles indicate SNPs significantly associated with the tested traits.

6,643 ddRAD SNP sites that passed filtering. From these, three SNPs were significantly associated with flight performance: two were associated with flight distance (Fig. 6A), and one associated with flight propensity (Fig. 6B). Of the two SNPs associated with flight distance, the first is located at position 3148638 on scaffold NW_017852012 (perm $p$ = 0.043), in an exon (XM_019902532) in gene LOC109536350 (Protein Brunelleschi). Beetles that are homozygous C ($n$ = 6) at this position are estimated to fly 1.06 km further than those that are heterozygous at this position ($n$ = 31), while those that are homozygous T ($n$ = 21) are estimated to fly 7.4 km less than heterozygous beetles. Although this SNP is in a coding region, it does not result in a change of amino acid. The second SNP associated with flight distance is located at position 786674 on scaffold NW_017852312 (perm $p$ = 0.045), in an intron of an uncharacterized gene, LOC109543163. Beetles that are homozygous A ($n$ = 44) at this position are estimated to fly 8.11 km less than those that are heterozygous at this position ($n$ = 14), and there were no beetles that were homozygous G at this site.

The SNP associated with flight propensity is located at position 188406 on scaffold NW_017851689 (perm $p$ = 0.046) and is located in an intron of the gene LOC109534071 (WD repeat-containing protein 55 homolog). Beetles that are homozygous T ($n$ = 2) at this position are estimated to initiate flight 2,173 more times than those that are heterozygous at this position ($n$ = 27), while those that are homozygous C ($n$ = 30) are estimated to initiate flight 68 fewer times than heterozygous beetles.

## DISCUSSION

Dispersal by flight is an important phase in the life history of many insects (*Roff & Fairbairn, 2007*). In the case of tree-killing bark beetles in their epidemic population phase, dispersal is mandatory for brood production (*Raffa, Grégoire & Lindgren, 2015*). Our study

shows that several genetic systems are associated with strong flight in the mountain pine beetle, with gene expression profiles of strong flying beetles being significantly different from both weak flying and non-flying beetles (Fig. 3); however, no significant differences were found between the weak fliers and non-fliers (Fig. 3), indicating that the physiological difference between non-fliers and weak fliers is minimal. Based on this, and in order to retain equal sample sizes, we focused on the differences between strong and weak fliers. Because we focus on associations, it is important to note that many of these genes may be a consequence of strong flight, rather than a contributor to it.

## Metabolism

Upregulated metabolic activity is expected for insect flight, as it is one of the most physiologically demanding activities known (*Wegener, 1996*). Several GO terms and KEGG pathways indicate the importance of metabolic processes, such as glycolysis, citric acid cycle, and serine metabolism. Serine, in particular, is associated with the metabolism of lipids (*Gao et al., 2018*), a known energy source in insects and a major fuel source for sustained flight in mountain pine beetles (*Evenden, Whitehouse & Sykes, 2014*; *Wijerathna & Evenden, 2019*). This is supported by several upregulated lipid metabolism-related genes and pathways (Table S2; Fig. 5).

## Muscle form & function

Differential expression of flight muscle-related genes affects flight capabilities in two armyworm species (*Spodoptera eridania* and *Spodoptera frugiperda*; *Portman et al., 2020*). We identified several differentially expressed genes that indicate the importance of muscle function and maintenance in flight in mountain pine beetle. We identified differentially expressed collagen genes, which are important for muscle development and function in monarch butterflies (*Danaus plexippus*; *Zhan et al., 2014*) and link to long distance flight behavior in insects such as the monarch butterfly (*Zhan et al., 2014*) and cotton bollworm (*Helicoverpa armigera*; *Jones et al., 2015*). Some myosins were also related to flight; in particular, paramyosin was upregulated in strong fliers, and is known to be structurally integral to thick filaments in the indirect flight muscle of *Drosophila* (*Drosophila melanogaster*; *Liu et al., 2003*). These genes may act in concert with various up and down regulated myotubularin genes in order to maintain muscle structure and function during bouts of sustained flight (*Laporte et al., 2001*).

## Oxidative stress and detoxification

Our data suggest several mechanisms are used by mountain pine beetles to mitigate stress-induced cellular damage during flight, including DNA repair pathways and GSTs. In strong fliers, DNA repair genes, supported by GO terms, are upregulated and would mitigate genomic damage caused by oxidative stress (Fig. 4). Also, GSTs are antioxidants (*Pompella et al., 2003*; *Yamamoto et al., 2011*; *Yamamoto et al., 2016*) and prevent build-up of potentially harmful byproducts of lipid metabolism in the flight muscle of *Drosophila* (*Singh et al., 2001*). Upregulated GSTs and other glutathione-related pathways in strong female fliers may play other roles in successful dispersal and establishment, as females are the

pioneering sex in the mountain pine beetle (*Blomquist et al., 2010*). For instance, GSTs are involved in detoxification of host defenses during colonization (*Keeling et al., 2013b*; *Robert et al., 2013*), and this process often involves precursor steps that rely on reactions with other enzymes, such as cytochromes P450 (*Sheehan et al., 2001*).

Several cytochromes P450 are differentially expressed in strong fliers, and these are found mostly in three cytochrome P450 families (CYP4, CYP6, and CYP9). These families are believed to be environmentally adaptive due to lineage-specific gene family expansions found in the mountain pine beetle when compared to *Tribolium castaneum* (*Keeling et al., 2013b*). Many of these are likely involved in host defense detoxification (*Sandstrom et al., 2006*; *Nadeau et al., 2017*; *Chiu, Keeling & Bohlmann, 2019a*). Others, in particular members of the CYP6 family, may be involved in pheromone biosynthesis (*Chiu, Keeling & Bohlmann, 2019b*), which may relate to variation in pheromone production observed in different mountain pine beetle flight phenotypes (*Jones et al., 2019*). While many CYP9 functions have not yet been characterized (*Keeling et al., 2013b*), we identified three differentially expressed CYP9 genes that were associated with strong flight. Other differentially expressed P450s in the CYP302 and CYP307 families may be related to the biosynthesis of hormones like 20E (*Iga & Kataoka, 2012*), which may impact dispersal-related behaviors, which is discussed in the following section.

## Endocrine system

Trends in gene expression in strong flying female mountain pine beetles indicate reduced levels of JH, and increases in 20E, insulin, and IGF. These are important hormones linked to dispersal and colonization in the mountain pine beetle. Reduced levels of JH in strong fliers may act to maintain dispersal by preventing the onset of several colonization-associated behaviors; during flight, JH epoxide hydrolase (JHEH), an enzyme that deactivates JH, is upregulated, likely reducing overall JH levels. In several insects, decreases in JH have been linked to dispersal (*Roff & Fairbairn, 2007*), however, increases in JH are linked to reproductive behavior (*Gruntenko & Rauschenbach, 2008*), biosynthesis of sex and aggregation pheromones (*Bridges, 1982*; *Conn et al., 1984*; *Hall et al., 2002*; *Tittiger et al., 2003*; *Tillman et al., 2004*), and degradation and reallocation of flight muscle resources (*Borden & Slater, 1969*; *Sahota, 1975*). Similar phenomena occur in the mountain pine beetle (*Mccambridge & Mata, 1969*) and may also be mediated by JH.

Increased interactions of insulin and insulin-like growth factors (IGF) were differentially expressed and may be associated with metabolism and growth. In many bark beetles, including the mountain pine beetle, strong dispersers tend to be larger (*Evenden, Whitehouse & Sykes, 2014*; *Shegelski, Evenden & Sperling, 2019*). This may be influenced by increased IGF receptors (*Nässel, Liu & Luo, 2015*) and reduced IGF binding protein levels (*Alic & Partridge, 2008*). Insulin, which shares structural similarities with IGF, affects carbohydrate and lipid metabolism (*Erion & Sehgal, 2013*) and is likely one of the metabolic hormones that increases metabolism in strong fliers.

We also identified upregulation of ecdysone 20-monooxygenase, which catalyzes the production of 20E (*Nigg et al., 1976*; *Johnson & Rees, 1977*) and may affect flight behavior

as it has links to lipid metabolism (*Wang et al., 2010*), reproduction (*Gruntenko & Rauschenbach, 2008*; *Sieber & Spradling, 2015*), and regulation of the circadian clock (*Kumar et al., 2014*).

**Flight behavior**

Several differentially expressed genes may mediate response to abiotic and biotic cues, affecting flight behavior. Strong fliers flew relatively more in the dark than weak fliers (Fig. 2) which may be linked to differential expression of light-response and circadian clock genes (Table S2). Nocturnin, which is downregulated in strong fliers, may affect response to environmental light cues, as it mediates circadian light responses in *Drosophila* (*Nagoshi et al., 2010*). Flight patterns in strong fliers also appeared to be more aligned with actual day and night (Fig. 2), which may be associated with differential response to a synchronized internal clock (*Wertman & Bleiker, 2019*). Also studied in *Drosophila*, circadian clock-controlled protein has circadian-controlled expression (*Lorenz, Hall & Rosbash, 1989*) and is upregulated in mountain pine beetles that are strong fliers. This, coupled with differential temporal flight performance, indicates potential differences in circadian synchronization that may affect dispersal capacity.

Differentially expressed olfaction genes may also be important in determining dispersal behavior. In the Douglas-fir beetle, *Dendroctonus pseudotsugae*, larger, stronger dispersers will often ignore chemical cues (*Bennett & Borden, 1971*), and this has similarly been observed in the mountain pine beetle (*Jones, 2019*). Several differentially expressed chemosensory proteins, odorant receptors, and odorant-binding proteins, most of which are downregulated in strong fliers, are potentially related to altered response to host volatiles, as occurs in another bark beetle, *Dendroctonus armandi* (*Zhang, Gao & Chen, 2016*), and these may be of interest in understanding the role chemical cues play in flight and dispersal behavior. Some chemosensory genes may influence flight behavior independent of chemosensory function (*Jones et al., 2015*); for example, although the mechanisms are not yet understood, odorant-binding proteins in *Helicoverpa armigera* have been linked to variation in flight activity, most likely through the use of lipids as flight fuel (*Wang et al., 2020*).

**Association study**

We identified three SNPs associated with dispersal capacity: two linked to total flight distance (Fig. 6A), and one linked to the number of times flight was initiated (Fig. 6B). Of these, one SNP associated with flight distance involved a synonymous change in an exon. The other two SNPs were in non-coding regions. Synonymous SNPs may influence genetic products (*Kimchi-Sarfaty et al., 2007*), and intronic SNPs can alter gene transcription regulatory elements (*Cooper, 2010*), but further research is needed to confirm causation and to investigate mechanisms that may be involved in variable flight performance. One of the genes containing a flight-associated SNP is uncharacterized. The other genes (protein Brunelleschi and WD repeat-containing protein 55 homolog) do

not currently have a known link to flight. Regardless, SNP associations with flight capacity could provide a novel monitoring tool for beetle management. Current mountain pine beetle surveying methods involve collecting disks from the surface of infested trees to determine reproductive success of a population. These disks often contain specimens that could be sequenced to detect markers of dispersal potential for that population based on flight-related SNP phenotypes.

## CONCLUSION

In conclusion, we identified genetic distinctions between mountain pine beetles with strong and weak flight capacity but found no significant difference between weak and non-fliers. The genetic systems associated with sustained flight are likely to be involved in meeting the physiological demands of flight, while downregulated systems may represent mechanisms to conserve resources for post-flight host colonization. Our work quantifies genetic coordination between dispersal and colonization. Our research also identifies genes that may contribute to dispersal potential through flight-related behaviors, as well as genetic variants associated with flight performance. These genes may be used to predict flight capacity in outbreak beetle populations through genetic testing, and they may also inform studies on the heritable nature of dispersal capacity in bark beetles, which has yet to be investigated. This study sheds light on the physiological nature of dispersal by flight and indicates several potential avenues for future research.

## ACKNOWLEDGEMENTS

The authors thank Julian Dupuis, Stephane Bordeleau, Jackson Lai, Dylan Sjølie, Phil Batista, Sebastian Lackey, Kelsey Jones, Antonia Musso, Devin Letourneau, Nathan Marculis, Melodie Kunegel-Lion, Janet Sperling and Stephen Trevoy for their help with sample collection, logistics, and advice on statistical analyses. This research was enabled in part by Westgrid and Compute Canada which both provided computation resources.

### Funding

This research was supported by funding awarded to Felix Sperling from the Natural Science and Engineering Research Council of Canada (grant no. NET GP 434810-12) to the TRIA Network, with contributions from Alberta Agriculture and Forestry, fRI Research, Manitoba Conservation and Water Stewardship, Natural Resources Canada–Canadian Forest Service, Northwest Territories Environment and Natural Resources, Ontario Ministry of Natural Resources and Forestry, Saskatchewan Ministry of Environment, West Fraser, and Weyerhaeuser. An NSERC Discovery Grant to F. Sperling (RGPIN-2018-04920) supported Victor A. Shegelski during preparation of this manuscript. The funders had no role in study design, data collection and analysis, decision to publish, or preparation of the manuscript.

## Grant Disclosures

The following grant information was disclosed by the authors:
Natural Science and Engineering Research Council of Canada: NET GP 434810-12.
TRIA Network.
NSERC Discovery Grant: RGPIN-2018-04920.

## Competing Interests

Dezene Huber is an Academic Editor for PeerJ. No other authors have competing interests.

## Author Contributions

- Victor A. Shegelski conceived and designed the experiments, performed the experiments, analyzed the data, prepared figures and/or tables, authored or reviewed drafts of the paper, and approved the final draft.
- Maya L. Evenden conceived and designed the experiments, authored or reviewed drafts of the paper, and approved the final draft.
- Dezene P.W. Huber conceived and designed the experiments, authored or reviewed drafts of the paper, and approved the final draft.
- Felix A.H. Sperling conceived and designed the experiments, authored or reviewed drafts of the paper, and approved the final draft.

## Field Study Permissions

The following information was supplied relating to field study approvals (*i.e.*, approving body and any reference numbers):

No field permit was required.

## DNA Deposition

The following information was supplied regarding the deposition of DNA sequences:

Raw RNA-Seq data files are available on NCBI SRA as FASTQ files (Accession # PRJNA664756.

Raw DNA data files are available on NCBI SRA as FASTQ files (Accession # PRJNA665051.

## Data Availability

A complete list of differentially expressed genes and the raw data (the distance flown and number of flight initiations data used for the association study, complete flight data for each specimen used in the differential gene expression study, and a summary of flight performed during the light and dark cycles of the bioassay) are available as Supplemental Files.

## Supplemental Information

Supplemental information for this article can be found online at http://dx.doi.org/10.7717/peerj.12382#supplemental-information.

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
