# Peer review of "Identification of genes and gene expression associated with dispersal capacity in the mountain pine beetle, Dendroctonus ponderosae Hopkins (Coleoptera: Curculionidae)"

_PeerJ, doi:10.7717/peerj.12382_

## Round 0.1 · original submission · Major Revisions

Dear Dr. Shegelski and colleagues:

Thanks for submitting your manuscript to PeerJ. I have now received two independent reviews of your work, and as you will see, the reviewers raised some relatively minor concerns about the research. This is great and indicates optimism for your work and the potential impact it will have on research studying the genetic factor underpinning dispersal capacity in the mountain pine beetle.

While the concerns of the reviewers are relatively minor, this is a major revision to ensure that the original reviewers have a chance to evaluate your responses to their concerns. There are many suggestions, which I am sure will greatly improve your manuscript once addressed.

Therefore, I am recommending that you revise your manuscript, accordingly, taking into account all of the issues raised by the reviewers. I do believe that your manuscript will be greatly improved once these issues are addressed.

Good luck with your revision,

-joe

Reviewer 1 ·

Basic reporting

The paper, “Identification of genes and gene expression…,” describes a study that uses transcriptomic and genomic methods to identify differential gene expression and SNPs associated with flight behavior in mountain pine beetle. Overall, the methods are robust and appropriate, and the paper provides valuable baseline information to guide future, more targeted studies, and perhaps will have applications for understanding beetle spread in response to the warming climate. I see no major flaws in the paper, and I commend the authors on their rigor and clear explanation of the study.

Experimental design

no comment

Validity of the findings

no comment

Additional comments

I have just a few suggestions that might improve the text, and some other comments, below.

Abstract:
Consider changing the wording of the second sentence. “Transcriptomics” and “genomics” are the methods, not the biological basis for dispersal. Maybe “gene expression” and “genetics,” or something similar, would fit better here. Also, in the second to last sentence, your study did not test whether beetles are conserving energy before host colonization, so this could be stated less conclusively.

Line 22: Do all of the papers in the block starting with “Robert et al.” address host defenses AND reproduction? Or should some be placed after “host defenses” and some after “reproduction”?

Line 72: Thank you for making sure the study organisms is documented with physical vouchers.

Line 268: Could delete “relatively.”

Line 289, and elsewhere: Some of the differentially expressed genes, like in this section, are not necessarily a trait of a “strong flier” or “weak flier”, but more likely a consequence of the activity of flight. It is a subtle difference, and probably splitting hairs, but I wonder if some slight wording changes could help make this distinction. As worded, I could see that future studies might cite your paper, portraying this incorrectly.

Line 305-308. The first sentence here and the last phrase in the second sentence seem redundant.

Conclusion:
The concluding paragraph is well-written, and accurately describes the value of the study.

Figures 3, 4 captions:
Please define “GO” and “KEGG” here again for readers who might not see them in the text.

Reviewer 2 ·

Basic reporting

English is unambiguous and professional. The writing is perfectly clear.

References are well presented. Maybe I would like to see a bit more detail on what the evidence is based that provides a possible functional link of a gene to dispersal capacity and in what organism.

Figures are clear.

ddRAD and raw expression data should be submitted to a public repository as well.

Experimental design

Very elegant experimental design with well discussed findings.

Validity of the findings

Some additional context and discussion could be helpful (see below).

Additional comments

Shegelski and colleagues present a study on genes and their expression associated with dispersal capacity in the mountain pine beetle D. ponderosae. They do this by first collecting beetles and testing their dispersal capacity (distance) and propensity (frequency of initiating flight). Next, they identified low and high dispersing individuals and used these for a gene expression and association study. They find a set of roughly 3,000 genes differentially expressed between dispersal phenotypes of which many present strong candidates for a functional link with dispersal metabolism or morphology. Using ddRAD for variant calling, they also identified 3 SNPs close to 3 genes that strongly associate with dispersal capacity and may be useful in the future to identify flight capacity at a population level.

I found the experimental setup of this study very elegant and the result interesting and well discussed. My main questions relate to elaborating some discussion. I think the analysis and interpretation are mostly strong.

First, I was wondering what the degree of phenotypic plasticity is of dispersal capacity and whether anything is known about the heritability of dispersal capacity. If heritability is low, few associations may be expected. It is mentioned that dispersal capacity may change after colonization, but I think it would be helpful to elaborate on this. If little is known, it would be interesting to the reader to mention this.

Second, in insects, cohorts with different dispersal capacity are often described as dispersal syndromes with discrete difference, because so many genes and traits are generally involved. Is this also the case for the mountain pine beetle, or is a continuum of phenotypes observed? This also raises the question if there might be key regulatory factors that give rise to the development of these phenotypes. For example, JH expression could be a key factor resulting in the development of a high or low dispersal morph, with downstream affects on thousands of genes. I think this would also be interesting to elaborate on, as it could be an eventual goal to identify these developmental switches or triggers.

Third, I felt a little confused about not presenting the RAD data for all loci. L240-253: What happens if you run this analysis on all SNPs in the genome? Adaptation in gene regulation may happen at a respectable distance from the actual genes and if the reference genome is highly fragmented, you might be losing many true positives.

Minor questions:

L92: Is the difference in genes because of the sex chromosome being absent in females?

L136-145: Does this methodology compare to a strategy followed by GEMMA (https://github.com/genetics-statistics/GEMMA), which allows for accounting for relatedness between individuals (which might be important if beetles come from the same tree)?

Figure1: might be more striking to present both A and B with the same y-scale. My first impression was that they don’t differ much. This is not crucial, but the results from L147-167 could maybe also be presented here (as box plots?)

L168-217: I like the summary of these result, but I am maybe somewhat lacking details on how these genes have been demonstrated to affect flight functions in other species. Also in the discussion, it would be interesting to know from what organisms this evidence comes.

L218: Please check consistency of writing e.g., 1,000 or 1000.

L240-241: How do you go from 714 to 367 scaffolds. Did the others not include any SNPs?

L241: How many ddRAD loci do these contain?

L300: “believed to be adaptive” Can you give a little more info on why this is?

---

## Round 0.2 · accepted · Accept

Dear Dr. Shegelski and colleagues:

Thanks for revising your manuscript based on the concerns raised by the reviewers. I now believe that your manuscript is suitable for publication without sending it back to the previous reviewers. Congratulations! I look forward to seeing this work in print, and I anticipate it being an important resource for groups on research studying the genetic factor underpinning dispersal capacity in the mountain pine beetle. Thanks again for choosing PeerJ to publish such important work.

Best,

-joe